# Reliability and agreement of a novel portable laser height metre

**Gustav Valentin Blichfeldt Sørensen** [1,2]*, **Johannes Riis**[1,2], **Mathias Brix Danielsen**[1,2], **Jesper Ryg**[3,4], **Tahir Masud**[5], **Stig Andersen**[1,2], **Martin Gronbech Jorgensen**[1]

1 Department of Geriatric Medicine, Aalborg University Hospital, Aalborg, Denmark, 2 Department of Clinical Medicine, Aalborg University, Aalborg, Denmark, 3 Department of Geriatric Medicine, Odense University Hospital, Odense, Denmark, 4 Department of Clinical Research, University of Southern Denmark, Odense, Denmark, 5 Department of Healthcare for Older People, Nottingham University Hospitals NHS Trust, Nottingham, England

* gustav.soerensen@rn.dk

## Abstract

### Background

Human height is a simple measure with great applicability. Usually, stadiometers are used to measure height accurately. However, these may be impractical to transport and expensive. Therefore, we developed a portable and low-cost laser height metre (LHM).

### Objective

We aimed to (1) determine intrarater and interrater reliability of our LHM and compare it to a wall-fixed stadiometer, (2) examine its agreement with the same stadiometer, and (3) determine the minimum number of recordings needed to obtain an accurate and reliable height measurement using the LHM.

### Methods

We recruited 32 participants (18+ years)–both men and women. Two raters performed assessments on the same day blinded to each other and their reference standard measurements. We calculated intraclass correlation coefficient (ICC), coefficient of variation (CV), standard error of measurement (SEM), and Bland-Altman plots with limits of agreement (LOA).

### Results

For both the LHM and stadiometer, we found ICC values of 0.99–1.00 (95% CI: 0.997–1.000) for both intrarater and interrater reliability. Regarding LHM intrarater reliability, SEM, CV, and LOA were 0.34 cm, 0.16%, and -1.07 to 0.73 cm, respectively. In terms of LHM interrater reliability, SEM, CV, and LOA were 0.27 cm, 0.12%, and -0.32 to 0.84 cm, respectively. As to agreement with stadiometers using one measurement, the mean difference was -0.14 cm and LOA ranged from -0.81 to 0.77 cm.

**Data Availability Statement:** All relevant data are within the paper and its Supporting Information files.

**Funding:** This study was funded and sponsored by the Department of Geriatric Medicine, Aalborg

University Hospital, Aalborg, Denmark, and
Department of Clinical Medicine, Aalborg
University, Aalborg, Denmark.

**Competing interests:** The authors have declared
that no competing interests exist.

## Conclusion

A portable and low-cost LHM, for measuring body height once, showed an excellent reproducibility within and between raters along with an acceptable agreement with a stadiometer thereby representing a suitable alternative.

## Introduction

Body height is used in many different contexts, e.g. when calculating body mass index [1] or reference intervals for normal lung function [2]. Also, height measures may be used when screening for vertebral fractures in patients with osteoporosis [3] and growth retardation in children [4] together with being used for baseline measurements in clinical studies. In a clinical setting, height is typically measured using a wall-fixed stadiometer, which serves as a costly golden standard [2,5,6]. However, newer studies have investigated alternative ways of measuring height using portable measuring devices with laser distance metres [7–9]. The advantage of these devices is the ability to perform reliable measures in alternating settings. Recently, researchers developed such a measuring device that required manual adjustments of measuring axes, which showed a systematic bias of 0.45 cm when compared to a stadiometer [7]. To optimise this, the authors suggested that future measuring devices using laser distance metres should be fixed in one or more measurement axes. Other researchers have developed such an instrument where the laser distance metre was fixated in two out of three measurement axes, which reduced the bias to 0.07 cm [9]. However, to measure the height using this fixated device, the distance from the skull to the ceiling has to be subtracted from the floor-to-ceiling distance, which is a more time-consuming procedure that potentially can introduce calculation errors. Therefore, we developed a portable laser height metre (LHM) using a laser distance metre fixated in two axes, which can produce height estimates without having to do post-measurement calculations. Thus, we aimed to determine intrarater and interrater reliability of the LHM along with comparing it to a stadiometer, examine its agreement with the stadiometer, and finally to determine the minimum number of recordings needed when performing a height measurement.

## Materials and methods

This paper is reported according to the Guidelines for Reporting Reliability and Agreements Studies (GRRAS) [10]. A protocol was registered a priori at Clinicaltrials.gov: NCT04070638.

### Participants

We recruited 32 participants (59% women) and performed measurements over two consecutive days, with 15 participants on the first day and 17 on the second day. The only inclusion criterium for the study was a minimum age of 18 years. Participants were students and colleagues at Aalborg University Hospital and Aalborg University recruited through convenience sampling, i.e. a non-consecutive manner, to maximise recruitment efforts during the time of the study. We collected data on gender, age, and height. No statistical sample size calculations were performed, but we decided to aim for the inclusion of 30 participants or above based on previous recommendations [11].

## Ethics

Written informed consent was obtained from all participants before participation in the study. The local ethics committee of Region Nordjylland, Denmark, was consulted, and approval of the study was not required for this study according to the Danish Act on the Scientific Ethical Committee System (Act no. 593, section 14, subsection 2). Approval was obtained from the Danish Data Protection Agency (record number 2019–100).

## Measuring devices

**Laser height metre.** A laser distance metre (Bosch Zamo, Robert Bosch GmbH, Gerlingen-Schillerhöhe, Germany) was mounted perpendicularly to the end of a wooden lath (3×4×50 cm) (Fig 1). The other end of the lath was mounted perpendicularly to a T-shaped metal plate thereby fixating the x- and z-axis of the LHM. To adjust the y-axis, a bubble level was mounted on the end of the lath to ensure that the LHM measured vertically down in its z-axis from the participants' vertex to the floor in front of the participants. The LHM weighed 800 g and was produced for 70 €. The device reports measures in meters to the thousandth's decimal equivalent to centimetres to the tenth's decimal.

**Stadiometer (reference standard).** Wall-fixed stadiometers were used as a reference (Harpenden Stadiometer, Holtain Limited, Crosswell, UK [6]). The stadiometers are calibrated once a week by a biomedical laboratory technician. The device reports measures in centimetres to the tenth's decimal.

## Procedure

Measurements were performed at Aalborg University Hospital, Aalborg, Denmark, from the 26th to 27th of August 2019 in two separate rooms next to each other with stadiometers installed. Two raters were conveniently chosen; a medical doctor and a medical student who both were researchers at the Department of Geriatric Medicine at the same hospital, but had no prior experience or training using either of the devices. They independently, i.e. without communicating with each other, performed height measurements on all participants in the two separate rooms with each measuring device, i.e. the LHM and stadiometer. Independence between raters was ensured by the first author observing both raters' behaviour simultaneously from outside the rooms. Each rater performed three measures per device. During measurements, participants were asked to stand flat on the floor–a hard surface–without shoes. Furthermore, participants were asked to stand with their heels positioned together and against the wall during both types of measurements. Each rater made sure that participants' heads were positioned in the Frankfurt plane, defined as the horizontal "plane passing through the upper periphery of the external auricular canals and the lowest point of the left orbit [12], and encouraged them to stand with a straight back against the wall. If participants had piled-up hair, they were asked to smooth this out to minimise any potential overestimation of height. Then for the stadiometer, the measuring plate was pulled down to the skull. Afterwards, the participants were asked to take a deep breath and hold it after which the measurement was performed. For the LHM, the device was placed on top of each participant's vertex and fixated in two axes by holding the T-shaped piece against the wall (Fig 2). The lath was rotated until the laser distance metre pointed vertically according to the bubble level at the end of the lath. The laser distance metre was activated by clicking on the "START" button. Participants were asked to take a deep breath and hold it, after which the measurement was locked and recorded. Following each measurement with both the stadiometer and LHM, the participant took a step away from the measuring area while the LHM was removed from the wall along with the position of the measuring plate of stadiometer was changed. Afterwards, the participant stepped

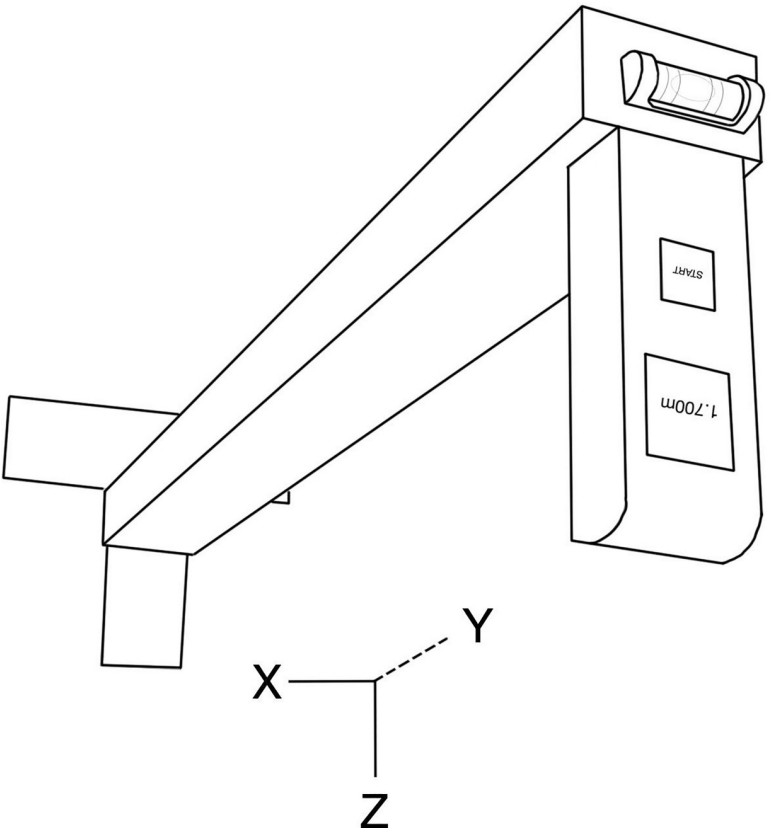

**Fig 1. The laser height metre.**

back again into the measuring area to replicate the measurement procedure. Measurements were recorded in centimetres to tenth's place since this was considered more clinically relevant than metres to the thousandth's place.

**Blinding.** The blinding process is illustrated in Fig 3. Thus, all participants were first measured by each rater using the LHM. When this stage was completed, measurements were archived and made inaccessible to blind the raters when performing stadiometer measurements again afterwards. All participants were measured in the same sequential order by both raters. The participants were not informed of their results of the height measurements and therefore blinded. We assumed that each rater was not able to remember each participant's height when all participants were measured consecutively. After the stadiometer measurements were performed, data were also archived and made inaccessible to raters. After 30 minutes, this entire procedure was repeated once more. Measurements were collected using a data capturing tool [13]. To guide readers, we defined a session as the process where one rater performed three measurements using one device, i.e. either the stadiometer or LHM.

**Statistics.** Histograms and QQ-plots were used to check for normal distributions. Age was reported in median and interquartile range due to a non-normal distribution that did not normalise when using logarithmic transformation, see S1 Appendix. Gender was reported in proportions. All heights measures were normally distributed and reported as mean (SD). For agreement measures, the mean of all three measurements for each participant per session was calculated. Afterwards, a group mean was calculated by taking the average of all participant means for each session. Subsequently, the difference between the group means was calculated

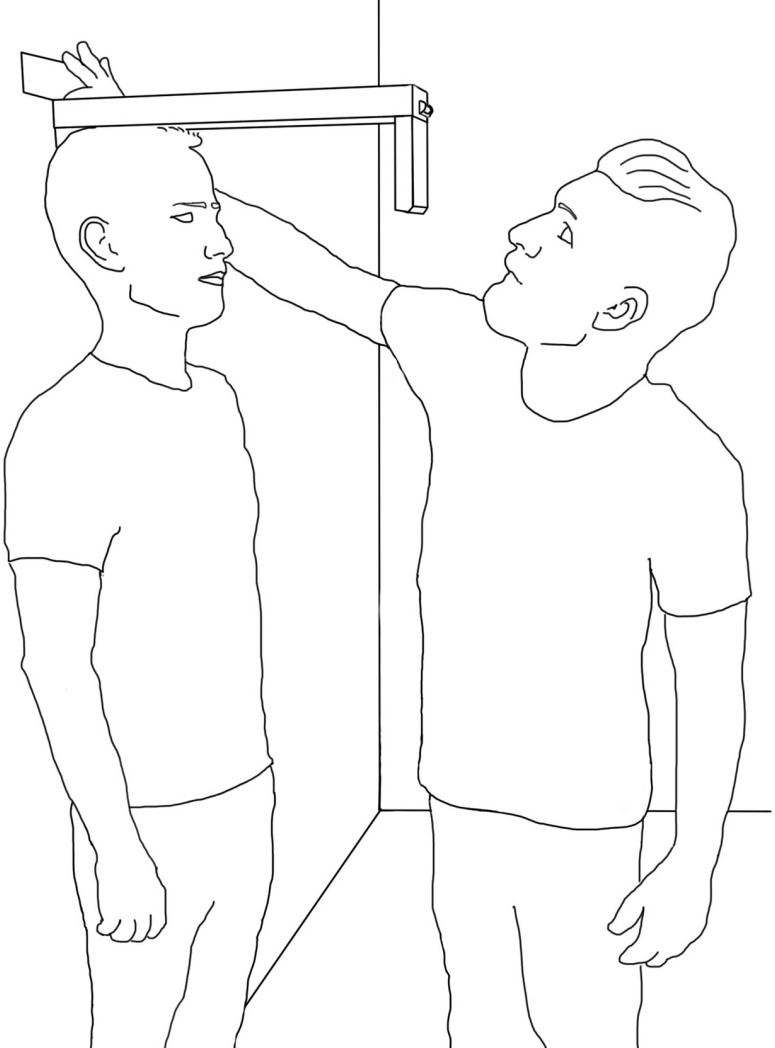

**Fig 2. Demonstration of how to use the laser height metre.**

by subtracting the LHM group mean from the stadiometer group mean for each rater. Furthermore, a two-tailed paired t-test was used for assessing systematic bias using the abovementioned difference. To evaluate whether using the average of three measures would have a similar agreement when using only two or one measurement, this process was also performed with the mean of the two first measurements, and finally exclusively the first measurement. We generated Bland-Altman plots with limits of agreement (LOA) for all the above-mentioned combinations. The same approach was used for intrarater and interrater reliability, where the difference in test-retest and between raters were investigated instead of between devices. LOA were calculated as mean difference $\pm 1.96 \times SD_{difference}$, where $SD_{difference}$ is the standard deviation of differences between two measurements on a sample of participants [14]. For the mean difference, 95% confidence intervals were calculated as mean difference $\pm 2.0395 \times SE_{bias}$, where 2.0395 is the t-value for 31 degrees of freedom (n-1) and a significance level of 0.05, and $SE_{bias}$ = $SD_{difference}/\sqrt{n}$ where n is the sample size. For LOA, 95% confidence intervals were calculated as LOA $\pm 2.0395 \times SE_{LOA}$, where $SE_{LOA} = 1.71 \times SE_{bias}$ [14]. For relative reliability measures, we calculated intraclass correlation coefficients (ICC) with 95% confidence intervals.

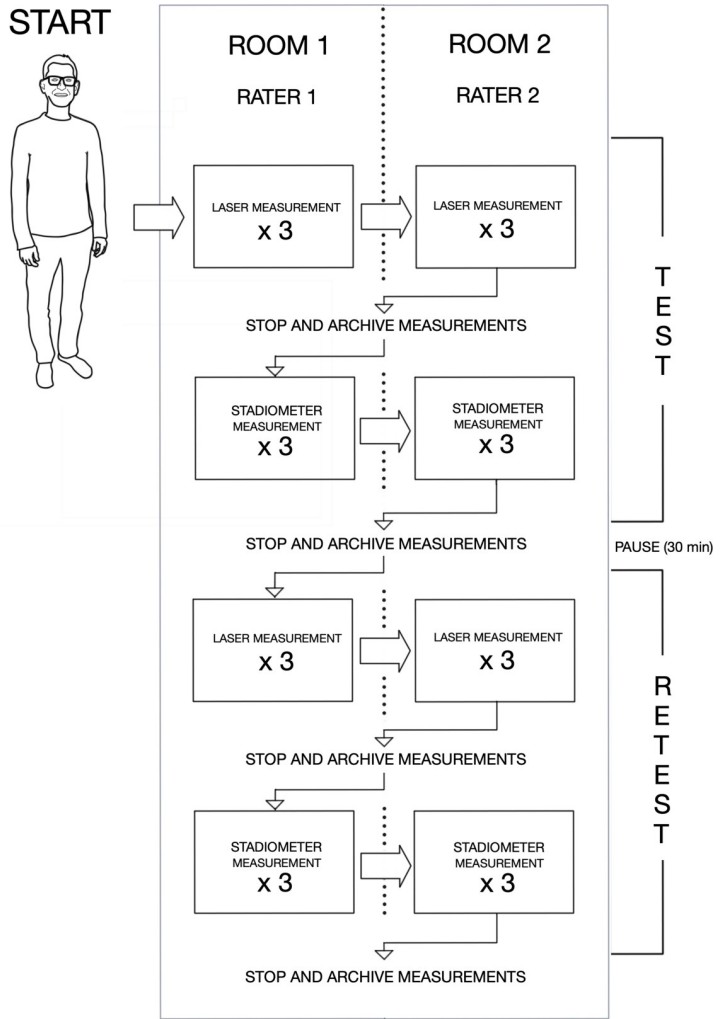

**Fig 3. Overview of the measuring sessions.**

ICCs for relative intrarater reliability were based on a single measurement, absolute agreement, two-way mixed effect model. For relative interrater reliability, we used a single measurement, absolute agreement, two-way random effect model (ICC 2,1) [11]. ICC estimates were interpreted as poor (<0.5), moderate (0.5–0.75), good (0.75–0.9), and excellent (>0.9) indicators of reliability [11]. For absolute reliability measures, we generated Bland-Altman plots, LOA, standard error of measurements (SEM), and coefficient of variation (CV) between raters on the same device (interrater) and as a test-retest setup (intrarater). SEM was calculated as the square root of the mean squared error term [15]. CV was calculated using the following formula CV(%) = $((\Sigma(SD/mean))/n)\times100$ [16]. Both reliability and agreement measures were analysed with a paired samples t-test to check for systematic bias and statistically significant differences, respectively [15]. Heteroscedasticity was inspected for using Bland-Altman plots. To be clinically applicable, we considered a difference in height measures less than 1 cm to be acceptable for both reliability and agreement measures. To avoid calculations based on typing errors, any identified observation on the measures differing more than 20 standard deviations from the remaining two measurements in that session were replaced with the mean of the same two measurements. The observation was replaced with the mean of the remaining two

measurements. Findings in the results section were reported conservatively by only displaying the most deviating measures. Thus, for each device, we reported the largest mean difference, lowest ICC, along with the highest CV, SEM and LOA measures found. Statistical analyses were performed using Microsoft Excel for Mac version 16.28 (Microsoft Office, Microsoft Corporation, WA), IBM SPSS Statistics for MacOS version 25.0 (IBM Corporation, Armonk, NY), and STATA for MacOS (StataCorp. 2019. Stata Statistical Software: Release 16. College Station, TX: StataCorp LLC).

## Results

Median age (IQR) was 23 (22–26) years. Mean (SD) height of participants was 175 (9) cm. Approximate time spent per participant was one minute for performing three measurements and recording them in the database. In the data set, one single outlier in the stadiometer measurements of 188.4 cm, equal 60 standard deviations from the remaining two measurements, was found. This was replaced with the mean of the two remaining measurements equal to 184.15cm. This has been highlighted in the complete and original data set that is fully available without any restrictions, see S2 Appendix. All analyses were performed on data from the 32 participants and reported in centimetres to the hundredth's place due to the statistical analyses summarising the measures originally reported in centimetres to the tenth's place.

### Intrarater reliability

Table 1 reports findings for intrarater reliability. For both the LHM and stadiometer, systematic bias for intrarater reliability was seen in both devices with -0.21 cm (95% CI: -0.33 cm to -0.07 cm; p = 0.003) and -0.14 cm (95% CI: -0.23 cm to -0.03 cm; p = 0.01), respectively. Both devices showed excellent relative intrarater reliability with ICC values of 0.99 (0.997–1.000). Also, absolute intrarater reliability was acceptable with a SEM, CV, and LOA of 0.34 cm, 0.16% and -1.07 (95% CI: -1.35 cm to -0.79 cm) to 0.73 cm (95% CI: 0.45 cm to 1.01 cm), respectively, for the LHM. This was 0.22 cm, 0.10%, and -0.69 (95% CI: -0.88 cm to -0.50 cm) to 0.53 cm (95% CI: 0.34 cm to 0.72 cm), respectively, for the stadiometer.

### Interrater reliability

Table 2 displays results for interrater reliability. No statistically significant systematic bias was seen in the LHM with a difference of 0.1 cm (95% CI: -0.04 cm to 0.24 cm; p = 0.15) between raters. However, a systematic bias of -0.14 cm (95% CI: -0.22 cm to -0.05 cm; p = 0.0026) was present for the stadiometer. For the LHM, SEM, CV, and LOA were 0.27 cm, 0.12%, and -0.64 (95% CI: -0.88 cm to -0.41 cm) to 0.84 cm (95% CI: 0.61 cm to 1.08 cm), respectively. Regarding the stadiometer, the same measures were 0.20 cm, 0.09%, and -0.63 (95% CI: -0.81 cm to -0.46 cm) to 0.47 cm (95% CI: 0.29 cm to 0.64 cm), respectively. Relative reliability was excellent with ICC values for both devices of 0.99 (95% CI: 0.998 to 1.000). When using the average of several LHM measurements, LOA narrowed and SEM along with CV decreased up to 0.08 cm and 0.03%, respectively, for both intrarater and interrater reliability.

### Agreement

Table 3 displays method comparison results. We found acceptable agreement regardless of the number of measurements used. When only measuring once, the mean difference was -0.14 cm (95% CI: -0.23 cm to -0.06 cm; p = 0.002) and LOA ranged from -0.81 (95% CI: -1.03 cm to -0.59 cm) to 0.77 cm (95% CI: 0.56 cm to 0.98 cm). Also, when using the average of three measurements, the mean difference was -0.15 cm (95% CI: -0.22 cm to -0.08 cm; p < 0.001) and

**Table 1. Intrarater reliability for height measurements on each rater with the laser height metre and stadiometer (n = 32).**

| Device | Rater | Combination of measurements | Test mean, cm | Retest mean, cm | Mean difference, cm (95% CI) | ICC (95% CI) | CV, % | Lower LOA, cm (95% CI) | Upper LOA, cm (95% CI) | SEM, cm |
|---|---|---|---|---|---|---|---|---|---|---|
| Laser height metre | Rater 1 | First measurement | 174.85 | 175.02 | -0.17 (-0.33; 0.00)* | 0.99 (0.997;0.999) | 0.16 | -1.04 (-1.32; -0.77) | 0.71 (0.43; 0.99) | 0.33 |
| | | Mean of first two measurements | 174.85 | 175.02 | -0.17 (-0.30; -0.03)* | 0.99 (0.998;1.000) | 0.14 | -0.90 (-1.14; -0.67) | 0.57 (0.34; 0.81) | 0.29 |
| | | Mean of all three measurements | 174.86 | 175.04 | -0.18 (-0.32; -0.04)* | 0.99 (0.997;1.000) | 0.15 | -0.93 (-1.17; -0.69) | 0.57 (0.34; 0.81) | 0.30 |
| | Rater 2 | First measurement | 174.77 | 174.94 | -0.17 (-0.33; 0.00)* | 0.99 (0.997;0.999) | 0.15 | -1.07 (-1.35; -0.79) | 0.73 (0.45; 1.01) | 0.34 |
| | | Mean of first two measurements | 174.77 | 174.96 | -0.19 (-0.33; -0.05)* | 0.99 (0.997;0.999) | 0.13 | -0.96 (-1.20; -0.72) | 0.57 (0.33; 0.81) | 0.30 |
| | | Mean of all three measurements | 174.76 | 174.97 | -0.21 (-0.33; -0.07)* | 0.99 (0.997;1.000) | 0.12 | -0.90 (-1.12; -0.68) | 0.50 (0.28; 0.72) | 0.29 |
| Stadiometer | Rater 1 | First measurement | 174.74 | 174.88 | -0.14 (-0.23; -0.03)* | 0.99 (0.999;1.000) | 0.09 | -0.67 (-0.84; -0.50) | 0.41 (0.24; 0.58) | 0.21 |
| | | Mean of first two measurements | 174.79 | 174.90 | -0.11 (-0.19; -0.02)* | 1.00 (0.999;1.000) | 0.08 | -0.58 (-0.73; -0.43) | 0.37 (0.22; 0.52) | 0.18 |
| | | Mean of all three measurements | 174.79 | 174.89 | -0.10 (-0.19; -0.01)* | 1.00 (0.999;1.000) | 0.09 | -0.60 (-0.76; -0.44) | 0.40 (0.25; 0.56) | 0.19 |
| | Rater 2 | First measurement | 174.88 | 174.96 | -0.08 (-0.19; 0.03) | 0.99 (0.999;1.000) | 0.10 | -0.69 (-0.88; -0.50) | 0.53 (0.34; 0.72) | 0.22 |
| | | Mean of first two measurements | 174.88 | 174.94 | -0.06 (-0.14; 0.03) | 1.00 (0.999;1.000) | 0.07 | -0.50 (-0.64; -0.36) | 0.39 (0.25; 0.53) | 0.16 |
| | | Mean of all three measurements | 174.88 | 174.94 | -0.06 (-0.13; 0.01) | 1.00 (0.999;1.000) | 0.07 | -0.45 (-0.57; -0.33) | 0.32 (0.20; 0.44) | 0.14 |

* = significant difference in a paired samples t-test ($p < 0.05$), cm = centimetres, ICC = Intraclass Correlation Coefficient based on a single measurement, absolute agreement two-way mixed effect model, CI = Confidence Interval, CV = Coefficient of Variation, SEM = Standard Error of Measurements, LOA = Limit of Agreement

LOA ranged from -0.71 (95% CI: -0.92 cm to -0.51 cm) to 0.76 cm (95% CI: 0.56 cm to 0.97 cm). To aid clinicians in interpreting these results, Table 4 gives examples of how to apply systematic bias, SEM, CV, and ICC in clinical practice. For Bland-Altman plots along with histograms and QQ-plots, see S3 Appendix. In summary, none of the Bland-Altman plots showed signs of heteroscedasticity.

## Discussion

This study aimed to determine intrarater and interrater reliability of the LHM along with comparing it to a stadiometer, examine its agreement with the stadiometer, and finally to determine the minimum number of recordings needed when performing a height measurement. We found excellent relative intrarater and interrater reliability for LHM along with an acceptable systematic bias, absolute reliability, and agreement. Furthermore, measuring the height only once was found adequate, as outcomes did not change substantially with more measures used. We consider the LHM a suitable alternative to stadiometers as differences are very small between devices. Also, the LHM has other advantages in terms of price and portability.

### Intrarater and interrater reliability

Three previous studies have examined height measuring devices with laser distance metres [7–9], all of which investigated intrarater reliability. Only one study [7] reported an interrater reliability ICC value. This was 0.991 (0.988–0.994) which was lower compared to our findings.

**Table 2. Interrater reliability for height measurements on each test session with the laser height metre and stadiometer (n = 32).**

| Device | Session | Combination of measurements | Rater 1 mean, cm | Rater 2 mean, cm | Mean difference, cm (95% CI) | ICC (95% CI) | CV, % | Lower LOA, cm (95% CI) | Upper LOA, cm (95% CI) | SEM, cm |
|---|---|---|---|---|---|---|---|---|---|---|
| Laser height metre | Test | First measurement | 174.85 | 174.77 | 0.08 (-0.05; 0.21) | 0.99 (0.998;1.000) | 0.11 | -0.64 (-0.86; -0.41) | 0.80 (0.57; 1.03) | 0.26 |
| | | Mean of first two measurements | 174.85 | 174.77 | 0.08 (-0.05; 0.22) | 0.99 (0.998;1.000) | 0.11 | -0.64 (-0.86; -0.41) | 0.80 (0.58; 1.03) | 0.26 |
| | | Mean of all three measurements | 174.86 | 174.76 | 0.10 (-0.04; 0.24) | 0.99 (0.998;1.000) | 0.12 | -0.64 (-0.88; -0.41) | 0.84 (0.61; 1.08) | 0.27 |
| | Retest | First measurement | 175.02 | 174.94 | 0.08 (-0.04; 0.20) | 0.99 (0.999;1.000) | 0.10 | -0.56 (-0.76; -0.36) | 0.72 (0.52; 0.92) | 0.23 |
| | | Mean of first two measurements | 175.02 | 174.96 | 0.06 (-0.03; 0.15) | 1.00 (0.999;1.000) | 0.08 | -0.43 (-0.58; -0.28) | 0.54 (0.39; 0.70) | 0.18 |
| | | Mean of all three measurements | 175.04 | 174.97 | 0.07 (0.00; 0.15)* | 1.00 (0.999;1.000) | 0.07 | -0.32 (-0.45; -0.20) | 0.47 (0.35; 0.60) | 0.15 |
| Stadiometer | Test | First measurement | 174.74 | 174.88 | -0.14 (-0.22; -0.05)* | 1.00 (0.999;1.000) | 0.09 | -0.59 (-0.73; -0.45) | 0.32 (0.18; 0.46) | 0.19 |
| | | Mean of first two measurements | 174.79 | 174.88 | -0.09 (-0.16; -0.02)* | 1.00 (0.999;1.000) | 0.07 | -0.45 (-0.57; -0.34) | 0.28 (0.16; 0.39) | 0.14 |
| | | Mean of all three measurements | 174.79 | 174.88 | -0.09 (-0.15; -0.02)* | 1.00 (0.999;1.000) | 0.07 | -0.45 (-0.56; -0.34) | 0.28 (0.16; 0.39) | 0.14 |
| | Retest | First measurement | 174.88 | 174.96 | -0.08 (-0.18; 0.02) | 0.99 (0.999;1.000) | 0.08 | -0.63 (-0.81; -0.46) | 0.47 (0.29; 0.64) | 0.20 |
| | | Mean of first two measurements | 174.90 | 174.94 | -0.04 (-0.13; 0.05) | 0.99 (0.999;1.000) | 0.07 | -0.54 (-0.69; -0.38) | 0.46 (0.30; 0.61) | 0.18 |
| | | Mean of all three measurements | 174.89 | 174.94 | -0.05 (-0.13; 0.03) | 1.00 (0.999;1.000) | 0.07 | -0.50 (-0.64; -0.36) | 0.39 (0.25; 0.53) | 0.16 |

* = significant difference in a paired samples t-test ($p < 0.05$), cm = centimetres, ICC = Intraclass Correlation Coefficient based on a single measurement, absolute agreement two-way random effects model, CI = Confidence Interval, CV = Coefficient of Variation, SEM = Standard Error of Measurements, LOA = Limit of Agreement

The other two studies reported findings on intrarater reliability using technical error of measurement [8,9]. This measure is not included in the reporting recommended according to GRASS guidelines and we did not include this [10]. Furthermore, none of the studies provided LOA, CV, or SEM values. This hampers comparison with our findings. Only one study investigated interrater reliability on LHMs [8] and they reported only technical error of measurement. In our study, relative reliability was excellent with ICC values similar to the reference standard for both intrarater and interrater reliability. Also, the absolute reliability was not as low as for the stadiometer, but we consider it clinically acceptable since SEM differed less than 1 cm between raters and sessions. Differences between the LHM and stadiometer may be due to the former being slightly more unstable in terms of fixation compared to the latter. Also, the y-axis of the LHM was adjusted using a bubble level which required subjective judgements when interpreted. This could have been optimised using a digital level.

## Agreement

Our findings are in line with prior studies with laser distance metres with mean differences of 0.07 to 0.45 cm, compared to our findings of 0.15 cm, when using the average of three measurements [7–9]. Furthermore, LOA ranged from -0.49 to 0.63 cm [9] to -3.3 to 2.8 cm [8] compared with our findings ranging from -0.71 to 0.76 cm. Reason for discrepancies may be due to inclusion of children by others [7,8], different reference standards used for comparison [7–9], and the lack of blinding [7–9]. Interestingly, the laser measuring device with fixated

**Table 3. Agreement between laser height metre and stadiometer on each rater and test session (n = 32).**

| Session | Rater | Combination of measurements | Stadiometer mean, cm | Laser height metre mean, cm | Mean difference, cm (95% CI) | Lower LOA, cm (95% CI) | Upper LOA, cm (95% CI) |
|---------|-------|-----------------------------|---------------------|------------------------------|-------------------------------|-------------------------|-------------------------|
| Test | Rater 1 | First measurement | 174.74 | 174.85 | -0.11 (-0.24; 0.02) | -0.81 (-1.03; -0.59) | 0.60 (0.38; 0.82) |
| | | Mean of first two measurements | 174.79 | 174.85 | -0.06 (-0.17; 0.04) | -0.64 (-0.83; -0.46) | 0.53 (0.34; 0.71) |
| | | Mean of all three measurements | 174.79 | 174.86 | -0.07 (-0.19; 0.05) | -0.71 (-0.92; -0.51) | 0.57 (0.37; 0.78) |
| | Rater 2 | First measurement | 174.88 | 174.77 | 0.11 (-0.01; 0.23) | -0.55 (-0.76; -0.34) | 0.77 (0.56; 0.98) |
| | | Mean of first two measurements | 174.88 | 174.77 | 0.11 (-0.01; 0.23) | -0.54 (-0.54; -0.33) | 0.76 (0.56; 0.97) |
| | | Mean of all three measurements | 174.88 | 174.76 | 0.11 (-0.01; 0.23) | -0.53 (-0.74; -0.33) | 0.76 (0.56; 0.97) |
| Retest | Rater 1 | First measurement | 174.88 | 175.02 | -0.14 (-0.23; -0.06)* | -0.60 (-0.75; -0.46) | 0.32 (0.18; 0.47) |
| | | Mean of first two measurements | 174.90 | 175.02 | -0.12 (-0.20; -0.04)* | -0.54 (-0.68; -0.41) | 0.30 (0.17; 0.43) |
| | | Mean of all three measurements | 174.89 | 175.04 | -0.15 (-0.22; -0.08)* | -0.55 (-0.68; -0.42) | 0.25 (0.12; 0.38) |
| | Rater 2 | First measurement | 174.96 | 174.94 | 0.02 (-0.08; 0.12) | -0.54 (-0.71; -0.36) | 0.58 (0.40; 0.75) |
| | | Mean of first two measurements | 174.94 | 174.96 | -0.02 (-0.11; 0.06) | -0.47 (-0.61; -0.33) | 0.43 (0.28; 0.57) |
| | | Mean of all three measurements | 174.94 | 174.97 | -0.02 (-0.10; 0.05) | -0.44 (-0.57; -0.31) | 0.39 (0.26; 0.52) |

* = significant difference in a paired samples t-test ($p < 0.05$), cm = centimetres, LOA = Limits of Agreement, CI = Confidence Interval

axes [9] differed less from the reference standard and produced clinically acceptable LOA compared to the two devices without fixation [7,8]. This emphasises the importance of stabilising the measuring device. We thus suggest future studies to investigate ways to enhance the fixation of the x-, z-, and y-axis of the LHM more accurately without compromising portability of the tool. This could be done by adding an adhesive material to the T-shaped base to enhance fixation to the wall.

## Portability, practicality, and price

Our LHM shows excellent reliability and acceptable agreement. Additional advantages include the device being portable in contrast to the wall-fixed stadiometer. This may be an advantage to field studies in alternating settings, e.g. participants' own homes. Also, it is possible to obtain a height measure within a few seconds which may ease implementation of LHM. Time spent was not reported by the previous studies [7–9]. This would have been relevant to compare since time spent may have differed between devices. For example, to generate a height measure in one of the studies, the distance from the skull to the ceiling had to be subtracted from the floor-to-ceiling distance [9]. Furthermore, this procedure had to be performed three times to finally obtain the average height of the participant. Thus, the feasibility in clinical practice and research of previous LHMs remains to be settled. Besides, our LHM is only a prototype. However, the production cost for the prototype was low at around 70 € compared to the sales price of a wall-fixed stadiometer of 1,269 € [6]. This may support the implementation of LHMs. Finally, commercialised portable stadiometers of similar price ranges as the LHM are available

**Table 4. How should clinicians interpret our results?.**

| Parameter and estimate | Examples and explanations |
|---|---|
| **Systematic bias**<br>Example: 0.2 cm | Intrarater: If a subject is measured two times by the same rater over a period of time, a change of $\leq 0.2$ cm can be due to bias and can be adjusted for.<br>Interrater: If a subject is measured one time by two raters, a difference of $\leq 0.2$ cm can be due to bias and can be adjusted for. |
| **Limits of agreement (LOA)**<br>Intrarater and interrater reliability<br>Example: -0.32 to 0.84 cm<br>Method comparison<br>Example: -0.81 to 0.77 cm | Intrarater and interrater reliability: If we measure height once, we can expect that the next value will lie within the LOA with 95% certainty. Thus, for a LOA of -0.32 to 0.84 cm, if one rater measures a participant with the height 180 cm, we would assume the height found by the second rater, or the same rater in a retest session, will be somewhere between 179.68 cm (180–0.32 cm) to 180.84 cm (180+0.84 cm) with 95% certainty.<br>Method comparison: If we measure the height of a participant with a new method and compare it to a reference standard, we can expect that 95% of the differences between the methods lies from -0.81 to 0.77 cm. Thus, a participant with a height of 180 cm measured on the reference standard would lie between 179.19cm (180–0.81 cm) and 180.77cm (180+0.77 cm) on the new method. |
| **Standard Error of Measurement (SEM)**<br>Example: 0.30 cm | A SEM of 0.30 cm means that for a given height measure, the true height of the individual will lie within $\pm 1.96 \times 0.3$ cm = $\pm 0.588$ cm of the obtained height from the measurement. Thus, for a person with a height of 180 cm measured once, a second measurement will have to be above 180.588 cm (180+0.588cm) or below 179.412 cm (180–0.588 cm) to be a real change in height. |
| **Coefficient of variation (CV)** | CV is a unitless indicator on how much the measuring device varies from the first to the second measurement. Thus, a lower CV equals a more reliable method. Since the indicator is unitless, comparison can be made between measuring devices using different measuring scales, e.g. centimetres and inches. |
| **Intraclass correlation coefficient (ICC)** | A high ICC value means that there is a low random and systematic measurement error, and thus high relative reliability, when measuring on the same subject several times (intrarater) or by two raters (interrater). |

[17]. However, during our literature search in MEDLINE, we did not find other validation or reliability studies on portable stadiometers not already mentioned in this paper. Thus, even though alternative products are available, the scientific transparency may seem limited on these devices thereby hindering device comparison. Thus, we consider our LHM a portable, practical, and low-cost solution.

## Strengths and limitations

First, we examined both reproducibility and agreement along with reporting our study according to guidelines [10]. Second, we used a blinded setup with both raters blinded to each other and their measurement on the reference standard. We cannot rule out that a Hawthorne effect affected the raters, i.e. they may have tried to perfect their measurements due to an awareness of being observed since both knew their ratings would be compared to each other [18]. However, the time spent of approximately one minute for performing and recording three measurements may have hindered the raters from accomplishing this. Thus, we believe such an effect to be limited. Future studies could overcome this by blinding the raters to the display on the measuring devices, after which the reading may be seen by a second person who records it in a database. Also, this may reduce the risk of typing errors. As mentioned earlier, a single observation had a difference of 4 cm from the other two observations within that session. When correcting this, the reliability and agreement improved substantially for the stadiometer, but not the LHM. Thus, even though correction of the data set was performed, we consider this as not having influenced our results of the LHM. Third, we did not perform sample size calculations. However, in retrospect, this is less relevant since the ability to detect a clinically significant change, i.e. less than 1 cm, was seen. Fourth, given the participants' age and height

ranges, the findings of this study may have limited generalisability to populations outside of these ranges, e.g. a paediatric population. Future studies could overcome this by including a larger and more diverse sample of participants. Fifth, all participants were measured in the same sequential order by both raters, rather than a randomised order, thereby potentially introducing a risk of systematic bias in the measurements. However, we found a clinically irrelevant difference between devices of 0.14 centimetres for which reason the participants or raters had to systematically adjust their height or measuring devices with this length for each session. Even though in theory, this may be possible, raters and participants were blinded to prior measurements thereby compromising the opportunity to systematically adjust these measures. Thus, we consider this potential bias as only being able to influence our results to a minor degree. A final limitation is the need for a vertical wall for obtaining similar results found in this study. This could be overcome by mounting a second bubble level along the side of the lath of the LHM.

## Conclusion

Our findings combined with prior studies on height measuring devices with laser distance metres have shown great potential for accurate and reliable measures of height. Previous studies used technologies with different practical and methodological limitations. This study improves on these limitations by having developed a portable, quick and low-cost measure for human height that can provide reliable and accurate readings using only one measurement with a performance that compared to a stadiometer. The ability of our device, and others, to fixate two out of three measuring axes may have improved the reliability and agreement. This suggests a need for further research in ways to stabilise the devices in all three measuring axes.

## Supporting information

**S1 Appendix. Histograms and QQ-plots to check for normality.**
(PDF)

**S2 Appendix. Complete and original data set.**
(XLSX)

**S3 Appendix. Bland-Altman plots for intrarater and interrater reliability along with method comparison.**
(PDF)

## Acknowledgments

The authors would like to thank the participants for taking part in the study along with the Osteoporosis Outpatient Clinic at Aalborg University Hospital for letting us use their facilities and stadiometers.

## Author Contributions

**Conceptualization:** Gustav Valentin Blichfeldt Sørensen.

**Data curation:** Gustav Valentin Blichfeldt Sørensen.

**Formal analysis:** Gustav Valentin Blichfeldt Sørensen.

**Funding acquisition:** Stig Andersen.

**Investigation:** Gustav Valentin Blichfeldt Sørensen, Johannes Riis, Mathias Brix Danielsen.

**Methodology:** Gustav Valentin Blichfeldt Sørensen, Stig Andersen, Martin Gronbech Jorgensen.

**Project administration:** Gustav Valentin Blichfeldt Sørensen.

**Resources:** Gustav Valentin Blichfeldt Sørensen, Stig Andersen.

**Supervision:** Jesper Ryg, Tahir Masud, Stig Andersen, Martin Gronbech Jorgensen.

**Visualization:** Gustav Valentin Blichfeldt Sørensen.

**Writing – original draft:** Gustav Valentin Blichfeldt Sørensen.

**Writing – review & editing:** Gustav Valentin Blichfeldt Sørensen, Johannes Riis, Mathias Brix Danielsen, Jesper Ryg, Tahir Masud, Stig Andersen, Martin Gronbech Jorgensen.

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
