## [Decision Letter · Decision Letter 0]

26 Feb 2020

PONE-D-20-02226

Intrarater and interrater reliability and agreement of a novel portable laser height metre

PLOS ONE

Dear Dr. Sørensen,

Thank you for submitting your manuscript to PLOS ONE. After careful consideration, we feel that it has merit but does not fully meet PLOS ONE’s publication criteria as it currently stands. Therefore, we invite you to submit a revised version of the manuscript that addresses the points raised during the review process.

We would appreciate receiving your revised manuscript by Apr 11 2020 11:59PM. To enhance the reproducibility of your results, we recommend that if applicable you deposit your laboratory protocols in protocols.io, where a protocol can be assigned its own identifier (DOI) such that it can be cited independently in the future. For instructions see: http://journals.plos.org/plosone/s/submission-guidelines#loc-laboratory-protocols

We look forward to receiving your revised manuscript.

Kind regards,

Jason S Ng, OD, PhD

Academic Editor

PLOS ONE

Additional Editor Comments (if provided):

Both reviewers found the study useful and interesting. The reviewers did not find significant faults with the study and have found minor revisions that should be investigated and responses given. The study is overall well planned with important issues being addressed a priori and the manuscript is well written with good attention to detail as well as very useful tables and figures.

Further comments include:

1. One important limitation of this study is that all subjects were young adults and with a limited age range (22-26) and a limited height range (175cm/SD = 9cm). Thus, the generalizability to a pediatric population or otherwise is lacking. This should be addressed. Given how seemingly simple the data collection was, the sample size could have been much larger and a more diverse sample of subjects recruited.

2. Portable stadiometers of similar price are actually available, though they are not quite as portable as the LHM – however, some of these do not even require a wall: Hopkins road rod stadiometer. The authors should further address that portable/affordable stadiometers do exist and comment on the place of the LHM with respect to this idea.

3. The resolution of the LHM is not clear. Figure 1 shows the meter reading to the thousandths place. All reported data is to the hundredth. Please discuss.

4. It is stated that ‘all participants were first measured…using the LHM” by both raters. Were these participants presented to both raters in the same sequential order each time? Or was there sequence of data collection randomized? This is not addressed and could introduce other effects of bias.

5. Eliminating data based on outliers (and replacing it with better data), while it is stated in the discussion that it likely had not effect, in order to ‘avoid calculations based on typing errors’ does not seem like good science and can potentially be a source of bias. Data on the repeatability of LHM device on known measured sources would be helpful here. Data entry into the REDCap system should have been simple enough and gives readers a sense of doubt if such simple data cannot be collected and recorded perfectly the first time.

6. The data cited in the text from table 1, does not seem perfectly accurate. For example, the -0.21 and -0.14 are from different categories so this is confusing. Did the authors select the largest mean difference per device? But isn’t this for intrarater reliability and not agreement between devices (Table 3). Further, the data discussing on lines 188-190 have similar questions. SEM seems to align with 0.15% and not 0.16%. Again the reason why those particular data points are chosen is not clear to the reader. The data cited in the text on lines 192-193 from Table 2 also suffers from this lack of clarity on why/how that particular data point was chosen among others to make the point. The data cited on line 196 does not align with the cited CV. The same issue arises from data cited on lines 202-205 and table 3.

Journal Requirements:

2. Please include a caption for Figure 1.

Reviewers' comments:

Reviewer's Responses to Questions

**Comments to the Author**

1. Is the manuscript technically sound, and do the data support the conclusions?

Reviewer #1: Yes

Reviewer #2: Yes

2. Has the statistical analysis been performed appropriately and rigorously? 

Reviewer #1: Yes

Reviewer #2: Yes

3. Have the authors made all data underlying the findings in their manuscript fully available?

Reviewer #1: Yes

Reviewer #2: Yes

4. Is the manuscript presented in an intelligible fashion and written in standard English?

Reviewer #1: Yes

Reviewer #2: Yes

5. Review Comments to the Author

Reviewer #1: Overall evaluation

This is an interesting article describing the development of a laser height meter and tested the validity and reliability of the device. Usage of the device would enable accurate anthropometric measurement of the height. Authors may consider some minor changes as listed below.

Title

Authors may think making the title concise by omitting “intrarater and interarter”.

Abstract

Objective – in the second aim, the term “same stadiometer” is not clear. You may please revise it.

Main Text

Result section may be divided into different paragraph for a better presentation.

Conclusion may be paraphrased for better clarity.

Reference

Avoid old references wherever possible.

Reviewer #2: Dear Authors,

Thank you for your contribution to this field of study. Your article is well written and your data supports your conclusions. I am suggesting a minor edit:

Line 180 - Please edit "(9 cm)" it is unclear what this is associated with.

6. PLOS authors have the option to publish the peer review history of their article (what does this mean?). If published, this will include your full peer review and any attached files.

Reviewer #1: No

Reviewer #2: Yes: Ashley Bauman

---

## [Author Response · Author response to Decision Letter 0]

16 Mar 2020

Response to reviewers: PONE-D-20-02226

Additional Editor Comments:

1. One important limitation of this study is that all subjects were young adults and with a limited age range (22-26) and a limited height range (175cm/SD = 9cm). Thus, the generalizability to a pediatric population or otherwise is lacking. This should be addressed. Given how seemingly simple the data collection was, the sample size could have been much larger, and a more diverse sample of subjects recruited.

a. Answer: We acknowledge this, and it has now been emphasised in the “Strengths and limitations” section (line 300). It now specifically states: “Fourth, given the participants’ age and height ranges, the findings of this study may have limited generalisability to populations outside of these ranges, e.g. a paediatric population. Future studies could overcome this by including a larger and more diverse sample of participants.”.

2. Portable stadiometers of similar price are actually available, though they are not quite as portable as the LHM – however, some of these do not even require a wall: Hopkins road rod stadiometer. The authors should further address that portable/affordable stadiometers do exist and comment on the place of the LHM with respect to this idea.

a. Answer: We agree that portable stadiometers of similar price are available. However, during our literature search on scientifically validated instruments, we did not find further studies to support this point. Thus, the products are available but the validity of these has not been reported, and comparison between devices may not be possible. This information has been added to the “Portability, practicality, and price” section (line 279) which now specifically states: “Finally, commercialised portable stadiometers of similar price ranges as the LHM are available [17]. However, during our literature search in MEDLINE, we did not find other validation or reliability studies on portable stadiometers not already mentioned in this paper. Thus, even though alternative products are available, the scientific transparency may seem limited on these devices thereby hindering device comparison.”.

3. The resolution of the LHM is not clear. Figure 1 shows the meter reading to the thousandths place. All reported data is to the hundredth. Please discuss.

a. Answer: We apologise if this was unclear. The LHM reports measures in metres to the thousandths place, which is displayed in Figure 1, but not mentioned in the manuscript. In the S3 Appendix, the complete data set is reported in centimetres to the tenths place which was judged as more clinically relevant than meters to the thousandth's place. However, all tables report measures to the hundredth's place due to the statistical analyses summarising the measures. This has been added to the manuscript: 

i. First, the “Laser height metre” section (line 95) now reads: “The device reports measures in meters to the thousandth’s decimal equivalent to centimetres to the tenth’s decimal”. 

ii. Second, the “Stadiometer (reference standard)” section (line 100) now reads: “The device reports measures in centimetres to the tenth’s decimal.”. 

iii. Third, the “Procedure” section (line 127) now reads: “Measurements were recorded in centimetres to tenth’s place since this was considered more clinically relevant than metres to the thousandth’s place.”. 

iv. Fourth, the “Results” section (line 193) now reads: "All analyses were performed on data from the 32 participants and reported in centimetres to the hundredth's place due to the statistical analyses summarising the measures originally reported in centimetres to the tenth’s place.”.

4. It is stated that ‘all participants were first measured…using the LHM” by both raters. Were these participants presented to both raters in the same sequential order each time? Or was there sequence of data collection randomized? This is not addressed and could introduce other effects of bias.

a. Answer: We acknowledge this, and we cannot completely rule out a potential risk of systematic bias since participants were presented to both raters in the same sequential order each time. However, we consider this as insignificant due to the small difference found between measuring devices and the blinding of both participants and raters towards the measurements. This has been added to the manuscript: 

i. First, the “Blinding” section (line 133) now reads: “All participants were measured in the same sequential order by both raters. The participants were not informed of their results of the height measurements and therefore blinded.”.

ii. Second, the “Discussion” section (line 303) now reads: “Fifth, all participants were measured in the same sequential order by both raters, rather than a randomised order, thereby potentially introducing a risk of systematic bias in the measurements. However, we found a clinically irrelevant difference between devices of 0.14 centimetres for which reason the participants or raters had to systematically adjust their height or measuring devices with this length for each session. Even though in theory, this may be possible, raters and participants were blinded to prior measurements thereby compromising the opportunity to systematically adjust the measures. Thus, we consider this potential bias as only being able to influence our results to a minor degree.”.

5. Eliminating data based on outliers (and replacing it with better data), while it is stated in the discussion that it likely had not effect, in order to ‘avoid calculations based on typing errors’ does not seem like good science and can potentially be a source of bias. Data on the repeatability of LHM device on known measured sources would be helpful here. Data entry into the REDCap system should have been simple enough and gives readers a sense of doubt if such simple data cannot be collected and recorded perfectly the first time.

a. Answer: We agree that eliminating data based on outliers can be suboptimal in terms of good scientific practice and may be a source of bias. However, not being transparent about such a post-measurement correction may be even worse. Thus, we further highlighted the change in “Red” in the appendix with the original data set to increase transparency. We still believe this was a data entry error. Even though REDCap can be set up using a simple design, we decided not to put any minimum or maximum boundaries in the system when entering data, in order to detect any actual differences on several, i.e. in this case - more than one, measurements between devices. Finally, the repeatability results on the LHM remained the same, as the error only pertained to a single data entry on the stadiometer. The “Results” section (line 189) now reads: “In the data set, one single outlier in the stadiometer measurements of 188.4 cm, equal 60 standard deviations from the remaining two measurements, was found. This was replaced with the mean of the two remaining measurements equal to 184.15cm. This has been highlighted in the complete and original data set that is fully available without any restrictions, see S2 Appendix.”

6. The data cited in the text from table 1, does not seem perfectly accurate. For example, the -0.21 and -0.14 are from different categories so this is confusing. Did the authors select the largest mean difference per device? But isn’t this for intrarater reliability and not agreement between devices (Table 3). 

a. Answer: We apologise if this was unclear. As mentioned in the “Statistics” section (line 178), “findings in the results section were reported conservatively by only displaying the most deviating measures”. Instead of reporting all 36 mean differences for intra- and interrater reliability along with agreement measures, we chose to report only the largest mean difference per device in the manuscript. Also, to increase transparency, we reported all mean differences in the tables. The same approach was used for the SEM, CV, LOA and ICC measures. Thus, for the systematic bias in intrarater reliability measures (Table 1), mean differences ranged between -0.17 cm to -0.21 cm for the LHM and -0.06 cm to -0.14 cm for the stadiometer. Therefore, the most conservative measures reported in the manuscript (line 198) in this example would be -0.21 cm for the LHM and -0.14 cm for the stadiometer. For agreement between devices (Table 3), measures ranged between 0.02 cm to -0.14 cm when performing one measurement only. Thus, a difference of -0.14 cm was found to be the most conservative measure and was therefore reported in the manuscript (line 220). This has been clarified in the “Statistics” section (lines 178-180) which now reads: “Findings in the results section were reported conservatively by only displaying the most deviating measures. Thus, for each device, we reported the largest mean difference, lowest ICC, along with the highest CV, SEM and LOA measures found.”

7. Further, the data discussing on lines 188-190 have similar questions. SEM seems to align with 0.15% and not 0.16%. Again, the reason why those particular data points are chosen is not clear to the reader. 

a. Answer: We apologise if this was unclear. Line 200-202 (previously 188-190) reads: “Also, absolute intrarater reliability was acceptable with a SEM, CV, and LOA of 0.34 cm, 0.16% and -1.07 (95% CI: -1.35 cm to -0.79 cm) to 0.73 cm (95% CI: 0.45 cm to 1.01 cm), respectively, for the LHM“. From Table 1 it can be seen that the CV for the LHM ranged between 0.12% to 0.16%. The SEM ranged between 0.29 cm to 0.34 cm for which reason. Therefore, we chose to report the latter CV and SEM due to the same reason as mentioned in Item 6. 

8. The data cited in the text on lines 192-193 from Table 2 also suffers from this lack of clarity on why/how that particular data point was chosen among others to make the point. The data cited in line 196 does not align with the cited CV. The same issue arises from data cited on lines 202-205 and table 3.

a. Answer: We apologise if this was unclear. Lines 207-208 (previously lines 192-193) read: “No statistically significant systematic bias was seen in the LHM with a difference of 0.1 cm (95% CI: -0.04 cm to 0.24 cm; p = 0.15) between raters.”. Line 211 (previously line 196) reads: “Regarding the stadiometer, the same measures were 0.20 cm, 0.09%, and -0.63 (95% CI: -0.81 cm to -0.46 cm) to 0.47 cm (95% CI: 0.29 cm to 0.64 cm), respectively.”. Lines 220-222 (previously line 202-205) read: “When only measuring once, the mean difference was -0.14 cm (95% CI: -0.23 cm to -0.06 cm; p = 0.002) and LOA ranged from -0.81 (95% CI: -1.03 cm to -0.59 cm) to 0.77 cm (95% CI: 0.56 cm to 0.98 cm)”. These values correspond to the most conservative results displayed in Table 2 and Table 3 within each statistical measure, i.e. mean difference, ICC, CV, Lower LOA, Upper LOA, and SEM. The reasoning for this choice of reporting has been given in Item 6.

Reviewers’ comments:

1. Title: Authors may think of making the title concise by omitting “intrarater and interrater”.

a. Answer: We appreciate this and have now changed the title to “Reliability and agreement of a novel portable laser height metre”.

2. Main Text: Result section may be divided into different paragraph for a better presentation.

a. Answer: We acknowledge this and have now made a division in the results section that now includes the following subheadings: “Intrarater reliability”, “Interrater reliability”, and “Agreement”. 

3. Conclusion may be paraphrased for better clarity.

a. Answer: We recognise this and have now paraphrased the conclusion section (line 315) which now reads: “Our findings combined with prior studies on height measuring devices with laser distance metres have shown great potential for accurate and reliable measures of height. Previous studies used technologies with different practical and methodological limitations. This study improves on these limitations by having developed a portable, quick and low-cost measure for human height that can provide reliable and accurate readings using only one measurement with a performance that compared to a stadiometer. The ability of our device, and others, to fixate two out of three measuring axes may have improved the reliability and agreement. This suggests a need for further research in ways to stabilise the devices in all three measuring axes.”

4. Reference: Avoid old references wherever possible.

a. Answer: We agree that references should be based on the latest evidence. However, we find our references essential even though their publication date seems older.

5. Line 180 – Please edit “(9 cm)” it is unclear what this is associated with. 

a. Answer: We apologise if this was unclear. Line 187 (previously line 180) reads: “Mean (SD) height of participants was 175 cm. (9 cm).”. The “(9 cm)” is the standard deviation on the mean height of the participants. This has been edited and the sentence now reads: “Mean (SD) height of participants was 175 (9) cm”. This was reported according to the “Statistical Analyses and Methods in the Published Literature (SAMPL) Guidelines” recommended by the PLOS ONE submission guidelines.

---

## [Editor Report · Decision Letter 1]

25 Mar 2020

Reliability and agreement of a novel portable laser height metre

PONE-D-20-02226R1

Dear Dr. Sørensen,

We are pleased to inform you that your manuscript has been judged scientifically suitable for publication and will be formally accepted for publication once it complies with all outstanding technical requirements.

With kind regards,

Jason S Ng, OD, PhD

Academic Editor

PLOS ONE

Additional Editor Comments:

The authors have addressed the editors and authors comments adequately.

---

## [Editor Report · Acceptance letter]

27 Mar 2020

PONE-D-20-02226R1 

Reliability and agreement of a novel portable laser height metre 

Dear Dr. Sørensen:

I am pleased to inform you that your manuscript has been deemed suitable for publication in PLOS ONE. Congratulations! Your manuscript is now with our production department. 

With kind regards,

on behalf of

Dr. Jason S Ng 

Academic Editor

PLOS ONE